# Optical Characterization of Three Reference Dobsons in the ATMOZ Project – Verification of G.M.B. Dobson's Original Specifications

Ulf Köhler[1], Saulius Nevas[2], Glen McConville[3], Robert Evans[3], Marek Smid[4], Martin Stanek[5], Alberto Redondas[6], and Fritz Schönenborn[1]

[1]Met. Obs. Hohenpeissenberg, Deutscher Wetterdienst, Albin-Schwaiger-Weg 10, 82383 Hohenpeissenberg, Germany
[2]Physikalisch-Technische Bundesanstalt, Bundesallee 100, 38116 Braunschweig, Germany
[3]ESRL, NOAA, 325 Broadway, 325 Boulder, USA
[4]Czech Metrology Institute, Okruzni 31, 638 00 Brno, Czech Republic, (Dept. of Optics, Prague)
[5]Solar and Ozone Observatory, Czech Hydrometeorological Institute, Zamecek 456, 500 08 Hradec Kralove 8, Czech Republic
[6]Izaña Atmospheric Research Center, AEMET- Meteorological State Agency, C/ La Marina 20, 6 Planta, 38071 Santa Cruz de Tenerife, Spain

*Correspondence to*: Ulf Köhler (ulf.koehler@dwd.de)

**Abstract.** Three reference Dobsons (regional standards Dobsons No. 064, Germany and No. 074, Czech Republic as well as the world standard No. 083, USA) were optically characterized at the **P**hysikalisch-**T**echnische **B**undesanstalt (PTB) in Braunschweig in 2015 and at the **C**zech **M**etrology **I**nstitute (CMI) in Prague in 2016 within the EMRP ENV 059 project "Traceability for atmospheric total column ozone". Slit functions and the related parameters of the instruments were measured and compared with G. M. B. Dobson's specification in his handbook. All Dobsons show a predominantly good match of the slit functions and the peak (centroid) wavelengths with deviations between -0.11 and +0.12 nm and differences of the **F**ull **W**idth **H**alf **M**aximum (FWHM) between 0.13nm and 0.37nm compared to the nominal values at the shorter wavelengths. Slightly larger deviations of the FWHMs from the nominal Dobson data, up to 1.22 nm, can be seen at the longer wavelengths, especially for the slit function of the long D-wavelength. However, differences between the **E**ffective **A**bsorptions **C**oefficients (EACs) for ozone derived using Dobson's nominal values of the optical parameters on one hand and these measured values on the other hand are not too large in the case of both "old" **B**ass-**P**aur (BP) and "new" IUP-ozone absorption cross sections. Their inclusion in the calculation of the **T**otal **O**zone **C**olumn (TOC) leads to improvements of significantly less than ±1% at the AD- and between -1% and -2% at the CD-wavelengths pairs in the BP-scale. The effect on the TOC in the IUP-scale is somewhat larger at the AD-wavelengths, up to +1% (D074), and smaller at the CD-wavelengths pair, from -044% to -1.5%. Beside this positive effect gained from the data with higher metrological quality that is needed for trend analyses and satellite validation, it will be also possible to explain uncommon behaviours of field Dobsons during calibration services, especially when a newly developed transportable device TuPS from CMI proves its capability to provide similar results as the stationary setups in the laboratories of National Metrology Institutes. Then, the

field Dobsons can be optically characterised as well during regular calibration campaigns. A corresponding publication will be prepared using the results of TuPS-based measurements of more than 10 Dobsons in field campaigns in 2017.

## 1 Introduction

The first measurements of the TOC were started in the 1920s. Such observations became possible after the development of the Dobson spectrophotometer by G.M.B. Dobson (Dobson, 1931) at the University of Oxford. A small network of six stations (Oxford, Valentia, Lerwick, Abisko, Lindenberg and Arosa) was set up in 1926 (Dobson et al. 1927; Götz et al., 1934). The network grew slowly until the International Geophysical Year in 1957 (Dobson, 1968; Brönnimann et al., 2003) when a large global network for ground-based TOC observations based on the Dobson instruments was established and successfully operated. Up to 100 instruments were in operation by the end of the 1960s (Bojkov, 2010).

A detailed description of the physical basis and the derived algorithm to calculate TOC from measured raw data can be found in Dobson (1957a), Komhyr (1980) and Evans (2008). The essential "constants" used in the equations are the Extraterrestrial Constants (ETCs) and ozone absorption coefficients of each Dobson. The ETCs of reference Dobsons are independently determined in an absolute calibration procedure using the Langley plot method. An explanation of this absolute calibration method can be found in the above-mentioned manuals. Whereas the ETCs of field Dobsons are specific for each instrument and can be determined by regular intercomparisons with absolutely calibrated reference Dobsons, the absorption coefficients are assumed to be the same for all Dobsons. This assumption is based on the idea, that the optical alignments of individual Dobsons match the specifications in G.M.B. Dobson's manuals (Dobson, 1957a; Dobson, 1957b; Dobson, 1962). Although this simplification might be a significant error source for poorly aligned instruments, a better approach to avoid this shortcoming by using EACs has not been possible until now. The measurement of individual slit functions to determine the EACs being specific to each Dobson had been too complex and time-consuming in the past. Other simplifications like assumed constant effective temperature of the ozone layer, only latitudinally but not seasonally depending height of the ozone layer, etc. also contribute to the uncertainty of Dobson-derived TOC-values. All these error sources, however, are not so large and crucial, that the overall accuracy of TOC observations with well aligned and calibrated Dobsons are affected too much (see Basher, 1982).

In contrast to this simplification of using nominal absorption coefficients for all Dobsons, the more modern Brewer spectrophotometer, developed and introduced into the global network in the late 1970s and early 1980s, uses EACs, which are specific for each individual instrument and can be determined during the basic calibration procedure (Kerr et al., 1985). The EACs can be directly measured using special lamp tests during normal calibration services. Here, bandwidths and the centre wavelengths of the used wavelength bands are individually determined. The resulting slit functions are convolved with the corresponding ozone absorption cross-sections measured in the laboratory. Similar laboratory-based investigations

of Dobson instruments were first performed by Komhyr et al. (1993) and recently by Evans et al. (2012). In both cases the ozone cross-sections were used that are recommended by the International Ozone Commission (IO$_3$C) since 1992. These were measured by Bass and Paur (1985) and Paur and Bass (1985). As stated before the complex measurements of individual EACs in the laboratory were not possible to be regularly performed for a larger number of instruments. Thus, it has been assumed that each instrument's absorption coefficients agree with those of the world reference Dobson (Komhyr, 1989).

Intense and long-term comparisons between Dobson and co-located Brewer spectrophotometers in the past three decades revealed systematic differences between both types of instruments (Köhler, 1986; Köhler, 1988; Scarnato, 2010; Vanicek, 2006; Vanicek et al., 2012). One of the most important sources for these differences is the influence of the real "effective" temperature ($T_{eff}$) of the ozone layer on the ozone cross sections (larger at the Dobson wavelengths than at the Brewer wavelengths). This $T_{eff}$ represents the ozone-weighted mean of the stratospheric temperatures in the ozone layer. Its annual average used in the Dobson algorithms is -46.3$^o$ C. This value is assumed to be constant all over the year and independent of the stations' latitude and longitude, which is definitely not the case. Several publications refer to this effect and can explain a considerable amount of the annual oscillation of the Dobson-Brewer difference (Kerr, 1988; Kerr, 2002; Bernhard et al., 2005, Scarnato, 2009). Redondas et al. (2014) combined the influence of the temperature with different laboratory-determined ozone absorption cross-sections (Serdyuchenkov, 2013, University of Bremen, **I**nstitut für **U**mwelt**p**hysik, called IUP cross-sections) to show the effect, which results in an increasing difference between Dobsons and Brewers at decreasing temperatures. Dobson spectrophotometers measure approximately 1% lower TOC at a $T_{eff}$-drop of -10K.

The remaining differences between Dobson and Brewer instruments, but sometimes also between field and reference Dobsons have been partly traced back to uncharacterised instrumental features, e.g. imperfect alignment of the Dobson optics and resulting deviations from the nominal absorption coefficients according to G.M.B. Dobson's specifications (Bernhard et al., 2005). Hence, the direct optical characterization of the slit functions of the instruments will improve understanding of the remaining discrepancies and offer a metrological basis for improved TOC measurements. The EMRP ENV59 Project "Traceability for atmospheric total column ozone" (ATMOZ), which started in 2014, has offered opportunity to characterise the optical properties of several Dobson spectrophotometers. This work has been done in a close co-operation between National Metrology Institutes (NMIs) - the German PTB in Braunschweig and the Czech CMI in Prague - as well as partners from the Dobson network, such as DWD in Hohenpeißenberg, Germany, ESRL NOAA in Boulder, USA, CHMI in Hradec Kralove, Czech Republic, and Izaña Atmospheric Research Center, AEMET- Meteorological State Agency, Spain.

**2 Measurement procedures in the laboratories**

For the characterization of the reference Dobson instruments within the ATMOZ project, two different approaches were taken by the NMIs involved in this task. PTB used for the slit function measurements its spectrally tuneable laser facility

working in a nanosecond-pulsed mode. The advantage of this approach is an ample power of the laser beam available for the Dobson slit function measurements, intrinsically narrow bandpass and accurate monitoring of the laser wavelengths. The biggest challenge faced in this measurement approach was a nonlinear response of the Dobson PMT detectors to the pulsed laser radiation. The setup for the Dobson characterizations at CMI was based on a double-grating monochromator with an argon arc light source providing more radiation in the ultraviolet (UV) range than standard UV lamps. The biggest advantage of the CMI approach was the absence of the nonlinearity problem that was faced in the case of the laser-based measurements. The challenge to be solved here was the measurement of the Dobson detector signals with an appropriate signal-to-noise ratio. Both measurement setups used by the NMIs for the Dobson characterizations are presented in detail in the subsections below.

## 2.1 Measurement setup at PTB

The spectral characterizations of the reference instruments Dobson No. 083 and Dobson No. 064 at PTB were carried out at the PLACOS setup (Nevas et al., 2009) using an oscilloscope (Ojanen et al., 2012) as shown schematically in Figure 1. The laser system generates 6-7 ns pulses at 20 Hz repetition rate. The respective spectral bandpass is 5 cm$^{-1}$, which corresponds to FWHM values of < 0.05 nm in the UV spectral range. The laser wavelength was monitored by a wavemeter and a high-resolution spectrometer with a 0.1 nm bandpass and wavelength scale uncertainty of 0.01 nm. The laser beam was coupled via a liquid light guide into a 5 cm diameter integrating sphere. One output port of the sphere irradiated entrance diffuser of the Dobson instrument. Another port held a monitor photodiode. Currents from the anode of the Dobson PMT-detector and the monitor photodiode were fed via current-to-voltage converters into two channels of a fast-sampling oscilloscope. The time-resolved measurements by an oscilloscope allowed to minimize detrimental effects of the PMT-anode dark current and noise. Simultaneous measurements of both PMT and monitor detector signals by the oscilloscope were triggered by a synchronization signal from the laser system. The slit function was obtained by normalizing the quotient of the PMT and the monitor detector signals recorded as a function of the laser wavelength to the value at peak wavelength. The measurements were repeated using different PMT high voltage settings and laser power levels. Here, a nonlinear behavior of the Dobson PMT detectors under the short-pulse laser irradiation was observed. The apparent widths of the slit functions were dependent on the used laser power levels and the PMT voltage settings. To solve this problem, the nonlinearities were mapped out as a function of the two parameters (laser power and PMT voltage). Consequently, respective correction functions to account for the nonlinearity of the PMTs could be determined. They were then applied to the results of the slit function characterizations yielding consistent results for all the measurements. A more detailed description of the PTB characterizations can be found in Nevas et al., 2016.

## 2.2 Measurement setup at CMI

The experimental setup for laboratory-based characterization of the Czech reference instrument Dobson No. 074 is shown both in the schematic diagram presented in Figure 2 and with the photo presented in Figure 3. The core of the facility is a

double grating monochromator with reflective optics with the F number equal to #f/4,5 in Czerny-Turner subtractive mode configuration using a couple of ruled gratings 1200 g/mm, blazed at 250 nm. The input slit of the monochromator is illuminated by Maxi-Arc, an Argon plasma source of high UV intensity with spectrally monotonous shape in the spectral range 300 – 350 nm. A reflective optics system at the output slit side reduces the F number of the output beam down to #f/12 to fit the beam to the Dobson spectrometer's input optic. A flipping mirror turns the beam from horizontal to vertical plane leading it towards the entrance slit of the characterised Dobson spectrometer. About 10% of the beam is deflected by a splitter to a monitor detector, which allows to correct the time fluctuations and the wavelength dependence of the monochromator's output radiation. The wavelength scale of the monochromator was calibrated for a slit width of 0.1 nm FWHM with a method described in (White, Smid and Porovecchio, 2012). The uncertainty of the wavelength scale was ±0.015 nm. The characterization of the 6 slit settings of the Dobson spectrometer was done by scanning around the central nominal wavelengths in 0.1 nm steps. The scanned wavelength range was set for slit S2 (short wavelengths) to ± 2 nm around the central wavelength and ± 4 nm for the wider S3 slit (longer wavelengths), respectively. The optical output power level varied from 51 nW at 310 nm up to 62 nW at 340 nm. These measured signals were processed, which means that dark signal components were subtracted and corrections of light instability and wavelength dependence of the monochromator light output were applied. The measured slit functions were analysed for errors due to non-zero bandwidth of the measuring beam. It turned out that there was no need for any correction of the used 0.1 nm FWHM slit-width.

## 3 Results

### 3.1. Cross-sections, slit functions and effective absorption coefficients (EACs)

The derivation of the EACs for each individual Dobson using the specific slit functions $S(\lambda)$ measured in the laboratories is described in detail in Bernhard et al., 2005 and Redondas et al., 2014. For this calculation the following approximate Eq. (1) is used:

$$\text{Eq (1)} \quad \alpha_i = \frac{\int \sigma(\lambda)\, S_i(\lambda)\, \mathrm{d}\lambda}{\int S_i(\lambda) \mathrm{d}\lambda}$$

where

$\sigma(\lambda)$ is the ozone cross-section for the corresponding wavelength at the fixed temperature of -46.3° C for the Dobson network (after Bass and Paur since 1992 and after IUP in the future).

$S_i(\lambda)$ is the measured instrument slit function for the corresponding wavelength.

$\alpha_i$ is the approximated effective absorption coefficient EAC.

The above-mentioned cross-sections after Bass and Paur have been in use since 1992. However, the International Ozone Commission decided recently to replace these old cross-sections by new ones. After the first proposal to use the results

derived from **D**aumont, **B**rion and **M**alicet (DBM, Daumont et al., 1992, Brion et al., 1993 and Malicet et al., 1995) it was found by Redondas et al., 2014, that the IUP ozone cross-sections, determined at the University of Bremen (Gorshelev et al., 2013; Serdyuchenko, 2013), give a much better agreement between the TOC measured by Dobsons and Brewers. The introduction of these IUP cross-sections into the global network is finally decided, but not completed yet.

To get a complete picture of the impact of using the effective ozone absorption coefficients, it was decided to compare not only the various sets (nominal and effective ones) of coefficients after Bass and Paur, but also to include the TOC-values in this comparison derived using individual Dobson EACs based on the new set of IUP absorption cross-sections. It is a very simple, almost direct correlation between the TOC values and the variation of the EACs, apparent when looking to the

10    general ozone calculation formula for the single wavelength pair:

$$\text{Eq (2)} \quad X = \frac{[N - (\beta - \beta')\frac{mp}{p_0} - (\delta - \delta')\sec(SZA)]}{(\alpha - \alpha')\mu} ,$$

where

15    $X$ = total amount of ozone expressed in Dobson Units (1 DU = $10^{-5}$ m pure ozone at STP), or in atmo-cm;

$$N = L_0 - L = \log(I_0 / I'_0) - \log(I / I')$$

$I_0$ and $I_0'$ = intensities outside the atmosphere of solar radiation at the short and long wavelengths, respectively;

$I$ and $I'$ = measured intensities of solar radiation at the short and long wavelengths, respectively;

$\beta$ and $\beta'$ = Rayleigh scattering coefficients of air at the short and long wavelengths, respectively;

20    $m$ = ratio of the actual and vertical paths of solar radiation through the atmosphere, taking into account refraction

and the earth's curvature: airmass;

$p$ = atmospheric pressure observed at the station;

$p_0$ = mean sea level pressure;

$\delta$ and $\delta'$ = scattering coefficients of aerosol particles at the short and long wavelengths, respectively;

25    $SZA$ = solar zenith angle  - angular zenith distance of the sun;

$\alpha$ and $\alpha'$ = absorption coefficients of ozone at the short and long wavelengths, respectively; either the nominal or the

effective ones (EACs);

$\mu$ = ratio of the actual and vertical paths of solar radiation through the ozone layer, the mean height of the ozone

layer being 22 km if not approximated by latitude of the station.

30

The first term in the equation 2 - $N/(\alpha - \alpha')\mu$ - is the dominant one, which primarily determines the TOC value $X$. Thus, a change of the absorption coefficient ($\alpha - \alpha'$), e.g. when EACs are determined and applied, modifies the TOC almost in the same order. As a simple rule of thumb, one can state: 1% smaller EACs provide 1% higher TOC.

## 3.2. Implications of the "new" effective absorption coefficients

The laboratory measurements at PTB and CMI provided instrument-specific wavelength settings and slit functions of the various bands for each Dobson instrument. As overview, the complete set of slit functions for all Dobsons are plotted in Figure 4a-c. Figures 5 and 6 show the measured slit functions of all Dobsons in detail for the short wavelengths A-S2 (305.5 nm) and D-S2 (317.5 nm), respectively. An example of the results for the wider long-wavelength functions is given in Figure 7, which represents A-S3 (325 nm) for all three Dobsons.

The slit functions of the three reference Dobson spectrophotometers show consistent patterns with good agreement of the wavelength settings for all wavelengths. However, they have quite different shapes as compared to the nominal slit functions, especially for the longer wavelengths (see figures 4a-c, 5, 6 and 7 and tables 1 and 2). The deviations of the wavelength settings vary from -0.11 nm (D074 at C-S2) to +0.12 nm (D064 at D-S2). Though, more than 50% of the wavelength deviations are less than ±0.05 nm, which is an indication of a good optical alignment matching Dobson's specifications. Special tests with an intentionally wrong setting of the Q-levers, which are used for the wavelengths selection (see corresponding descriptions in the relevant manuals Dobson, 1957a, Komhyr, 1980 and Evans, 2008), reveal that a misalignment by the accepted limit of 0.3° results in a 0.05 nm-deviation from the correct wavelength. The shapes of the slit functions, represented by the FWHMs, are very close to the ideal Dobson specifications in the short wavelength range of slit S2, namely A-S2 (figure 5) and C-S2 and slightly worse at D-S2 (figure 6). The FWHM differences are less than 0.2 nm in A and C and around 0.3 nm in D. In the longer wavelength range of slit S3 the slit functions are significantly wider than the nominal ones (figure 7 for D-S3 and table 2). The deviations vary between +0.62 nm (D074 at A-S3) and +1.22 nm (D083 at D-S3). Thus, it is expected that the individual EACs deviate significantly from the specified values. The ratios EAC/nominal in table 3 show values up to 0.972 (D064, D-pair), which results in a TOC difference of nearly +3%. Fortunately, when looking at the combined wavelengths pairs AD and CD in table 3 the resulting differences are much lower: -0.31% up to +0.559 for AD and between -1.961% and -1.060% for CD.

Finally, these individual slit functions at the observed wavelengths are convoluted with the designated new IUP ozone cross-sections and the former Bass and Paur (BP) values to provide EACs as described in section 3.1. These EACs for BP (table 3) and IUP (table 4) are compared with the nominal BP values (after Komhyr, 1993).

The largest effects on the TOC calculation can be seen when using the single wavelength pairs, especially the D-pair. When applying the IUP-EACs, the A and C ozone values are between 0.79% (C of D083) and 1.49% (A of D074) higher than the nominal BP TOCs. The larger deviations of the EACs are observed at the D-wavelength pairs result in much higher TOC differences, which are between 3.5% (D074) and 4.82% (D064). Fortunately, the majority of the regular TOC data, submitted to the WOUDC (**W**orld **O**zone and UV **D**ata **C**enter) in Toronto and used for scientific purposes like trend analyses and satellite validation, are based on the AD-wavelengths pairs. Only a minor data set originated from CD-observations during winter season at higher latitude station, when sun is too low for AD. The changes of these TOC-values are less than -2% for CD with EAC-BP (table 3, last column) and -1.5% with EAC-IUP (table 4, last line). The differences of the revised AD data are less than ±1% in both cases. These results can explain the principal difference between the original AD- and CD-TOC, which are observed when using nominal BP absorption coefficients. The introduction of IUP-based absorption coefficients, either the nominal ones using the specified Dobson slit functions or the EACs based on the measured slit functions, will provide a better agreement between AD- and CD-TOC. The pure cross-section effect using the nominal slit functions will be about +0.9% for AD and +0.6% for CD (Redondas et al., 2014), which results in a general reduction of the AD-CD difference for all Dobsons by about 0.3%. The effect of using measured EACs is strongly depending on each instrument's specific slit functions and can be larger than 1%.

This application of Dobson EAC-IUP and additionally considering that the use of the IUP cross-sections also reduces Brewer TOC by 0.5% (Redondas et al., 2014), the principal negative difference between TOC obtained from Dobson AD and from Brewer spectrophotometers (calibrated by the RBCC-E at the Izaña Atmospheric Research Center, AEMET, Tenerife) will be reduced as well (see submitted paper of Redondas et al., this special AMT issue, probably published in 2018).

In addition to the above mentioned, already submitted paper a more detailed presentation of the TOC-measurements of all three reference Dobsons during the ATMOZ-campaign at Izaña campaign in September 2016 will be given in a separate publication, which will be submitted to AMT in 2018.

## 4 Summary, conclusion and outlook

The investigations of three reference Dobsons (D083 and D064 at PTB and D074 at CMI) revealed, that the optical alignment and properties of these instruments indeed deviate from the specifications postulated by G.M.B. Dobson. These differences, however, are not so large, that the derived EACs at the AD and CD wavelengths pairs of the standard TOC observations would lead to considerably changed TOC values. Largest changes will occur in the TOC obtained using only single wavelength pairs, e.g. the D-TOC can be higher by around 4%. Fortunately, the regularly used AD-TOC values are affected less than ±1%. Thus, correspondingly re-evaluated TOC data sets of these three instruments will not change

significantly. Though, the observed differences among individual Dobsons and between Dobson and Brewer instruments will be reduced.

A large intercomparison campaign, organised under the auspices of the ATMOZ project, held at the Izaña Atmospheric Research Center on Tenerife in September 2016, provided a very good data base to confirm this prognosis of an improved Dobson-Brewer agreement. A detailed investigation of the results of this campaign will be published in a separate paper (Redondas, et al., this special AMT issue, probably published in 2018). In addition, the CMI Prague developed a portable system TuPS (Tuneable Portable Radiation Source) (Porrovecchio et al., 2017). This system has a potential to facilitate the optical characterization of Dobson in situ within the time schedule of an hour, without the need of time demanding transport and characterization of Dobson spectrometers in the metrology laboratories. If comparisons of results collected during special campaigns with those obtained at the laboratory facilities at PTB and CMI confirm its capability for reliable and sufficiently accurate characterizations, this TuPS will become a new and valuable tool for Dobson calibration centres and, thus, will help to improve the quality of the calibration services. Consequently, the quality of the Dobson TOC records in the data centres will be improved as well, which will increase the reliability of these data for their use in trend analyses and satellite validations.

A potential application of the knowledge about the real slit functions and effective absorption coefficients will be discussed in the Dobson community. The additional efforts and effects of such a two-point calibration (ETCs and EACs) are not quite clear yet.

**Acknowledgement**

This work has been supported by the European Metrology Research Programme (EMRP) within the joint research project EMRP ENV59 ATMOZ "Traceability for atmospheric total column ozone". The EMRP is jointly funded by the EMRP participating countries within EURAMET and the European Union.

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

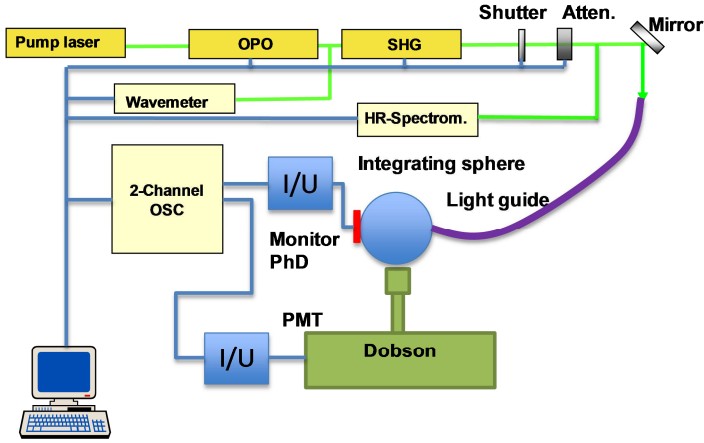

Figure 1: Schematic representation of spectral characterizations of
Dobson spectrophotometers at the PLACOS setup of PTB.
OPO: optical parametric oscillator; SHG: second harmonic generator;
OSC: oscilloscope; I/U: current-to-voltage coverter; PD: silicon photodiode.

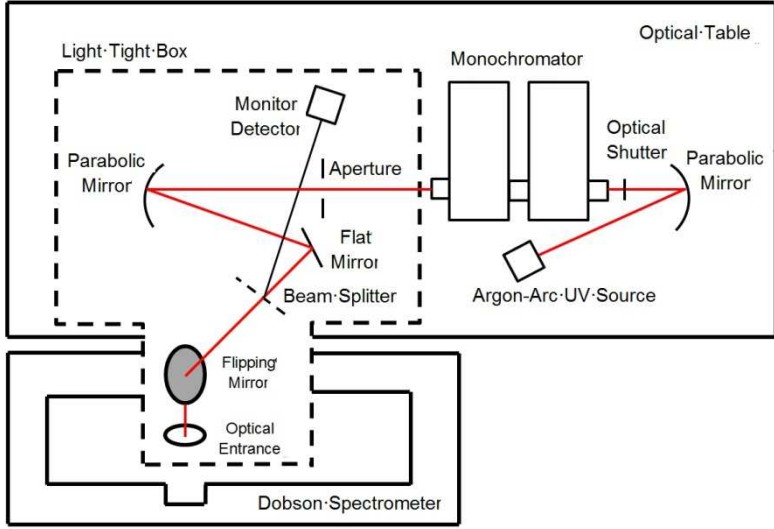

Figure 2: Schematic diagram of CMI monochromator-based facility used
for characterization of Dobson 074.

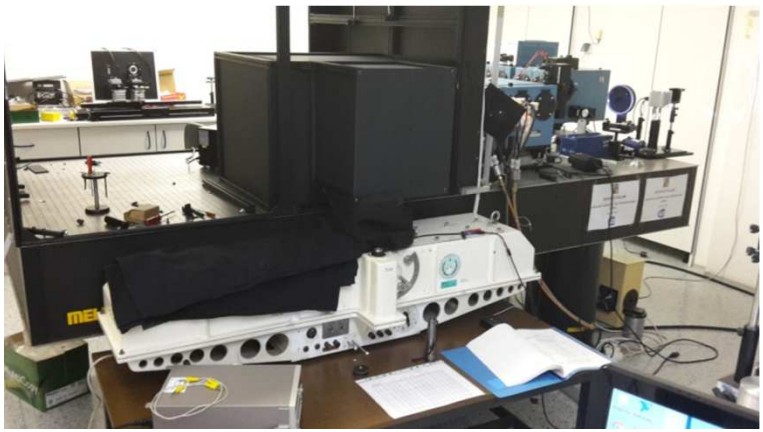

Figure 3: Photo of the CMI measurement setup. Double-grating monochromator is on the left side, the output optics light-tight box in the middle of the picture as well as the Dobson 074.

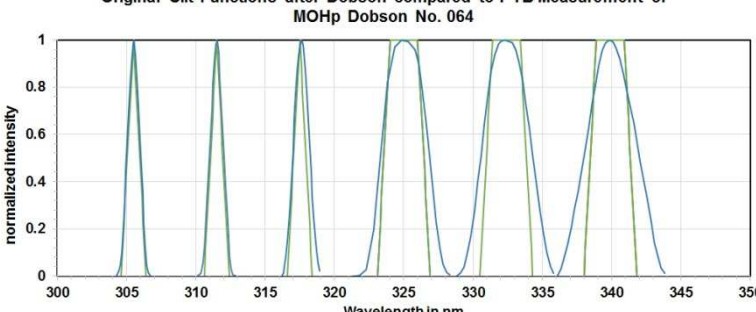

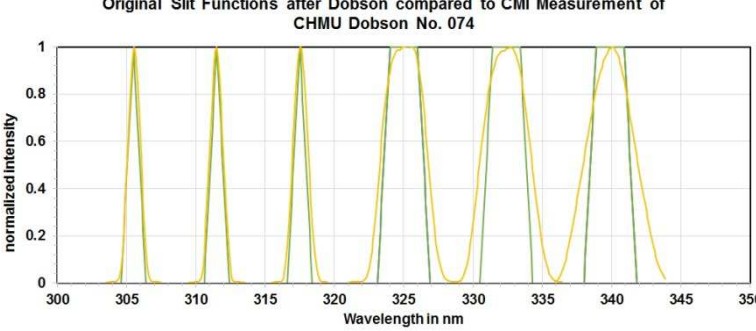

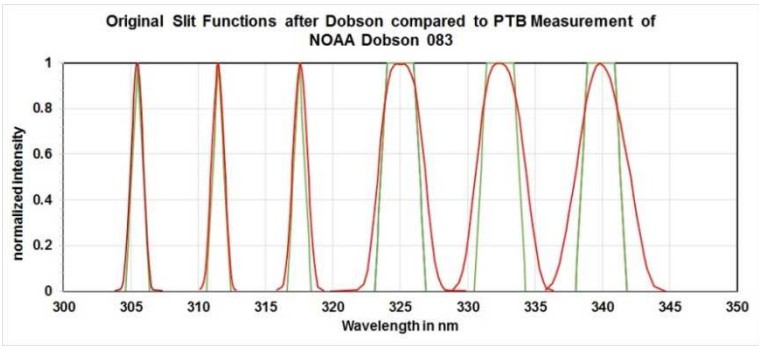

5    Figure 4a-c: All slit functions of all short (three curves on the left side)
and long wavelengths (three curves on the right side) for Dobsons No. 064 (a),
No. 074 (b), and No 083 (c) compared with nominal slit functions after Dobson.

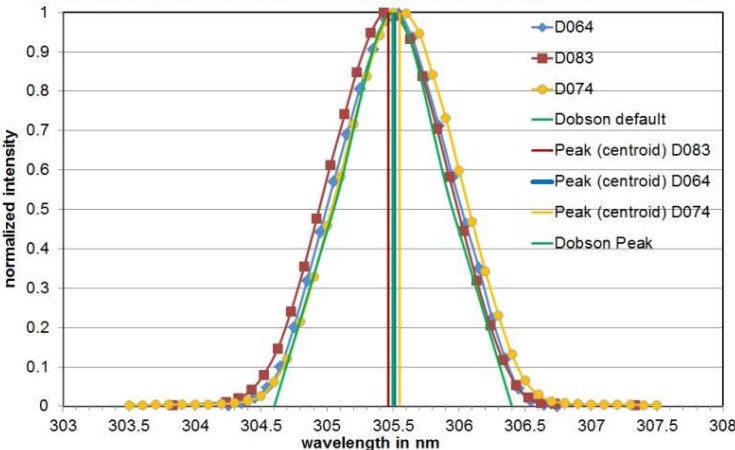

 Figure 5: Slit functions at the short wavelength
of wavelength pair A (A-S2) of all three reference Dobsons.

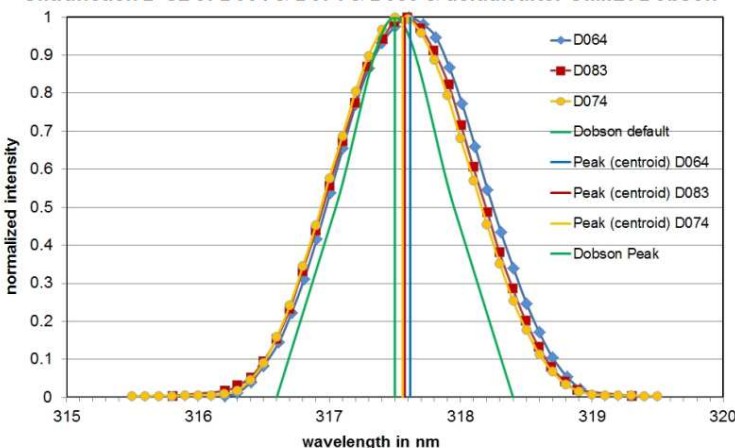

Figure 6: Slit functions at the short wavelength
of wavelength pair D (D-S2) of all three reference Dobsons.

Figure 7: Slit functions at the long wavelength
of wavelength pair A (A-S3) of all three reference Dobsons.

| | D083 (NOAA) | | D074 (CHMI) | | D064 (DWD) | |
|---|---|---|---|---|---|---|
| Slit/FWHM (nm) | Peak (nm) | FWHM (nm) | Peak (nm) | FWHM (nm) | Peak (nm) | FWHM (nm) |
| A-S2 (305.5/0.90) | 305.46 | 1.05 | 305.55 | 1.04 | 305.51 | 1.03 |
| C-S2 (311.5/0.90) | 311.47 | 1.09 | 311.49 | 1.09 | 311.50 | 1.08 |
| D-S2 (317.5/0.90) | 317.58 | 1.24 | 317.56 | 1.22 | 317.62 | 1.27 |
| A-S3 (325.0/2.90) | 325.10 | 3.56 | 325.05 | 3.52 | 325.08 | 3.56 |
| C-S3 (332.4/2.90) | 332.47 | 3.81 | 332.39 | 3.80 | 332.44 | 3.81 |
| D-S3 (339.9/2.90) | 340.00 | 4.12 | 339.94 | 3.98 | 339.97 | 4.06 |

Table 1: Measured centroid wavelengths (Peak) and FWHMs (Full Width at Half Maximum) for all Dobsons and wavelengths; nominal values are given in the first column in brackets.

| | D083 (NOAA) | | D074 (CHMI) | | D064 (DWD) | |
|---|---|---|---|---|---|---|
| Slit/FWHM (nm) | Peak (nm) | FWHM (nm) | Peak (nm) | FWHM (nm) | Peak (nm) | FWHM (nm) |
| A-S2 (305.5/0.90) | -0.04 | +0.15 | +0.05 | +0.14 | +0.01 | +0.13 |
| C-S2 (311.5/0.90) | -0.03 | +0.19 | -0.11 | +0.19 | +0.00 | +0.18 |
| D-S2 (317.5/0.90) | +.08 | +0.34 | +0.06 | +0.32 | +0.12 | +0.37 |
| A-S3 (325.0/2.90) | +.10 | +0.66 | +0.05 | +0.62 | +0.08 | +0.66 |
| C-S3 (332.4/2.90) | +0.07 | +0.91 | -0.01 | +0.90 | +0.04 | +0.91 |
| D-S3 (339.9/2.90) | +.10 | +1.22 | +0.04 | +1.08 | +0.07 | +1.16 |

5   Table 2: Measured differences to Dobson's specifications of wavelength settings and FWHMs; nominal values are given in the first column in brackets.

| Effect of Dobson characteristics measurements within ATMOZ | | | | | |
|---|---|---|---|---|---|
| **Absorption coefficients** | **A** | **C** | **D** | **AD** | **CD** |
| **Dobson/Komhyr (nominal)** | 1.806 | 0.833 | 0.374 | 1.432 | 0.459 |
| **D064** | | | | | |
| **Ratio EAC/nominal** | 0.997 | 0.993 | 0.972 | 1.003 | 1.011 |
| **EAC** | 1.800 | 0.828 | 0.364 | 1.436 | 0.464 |
| **Relative difference in % TOC** | | | | -0.310 | -1.060 |
| **D074** | | | | | |
| **Ratio EAC/nominal** | 0.993 | 1.006 | 0.989 | 0.994 | 1.020 |
| **EAC** | 1.794 | 0.838 | 0.370 | 1.424 | 0.468 |
| **Relative difference in % TOC** | | | | 0.559 | -1.961 |
| **D083** | | | | | |
| **Ratio EAC/nominal** | 0.997 | 0.998 | 0.983 | 1.001 | 1.011 |
| **EAC** | 1.800 | 0.832 | 0.367 | 1.433 | 0.464 |
| **Relative difference in % TOC** | | | | -0.060 | -1.120 |
| **2010-results** | 1.805 | 0.830 | 0.376 | 1.429 | 0.454 |

Table 3: Effective Absorption Coefficients (EACs) calculated with Bass/Paur cross-sections, their ratio to the nominal ones and the resulting relative difference in % TOC. 2010-results from measurements of the D083 spectral characteristics (Evans et al., 2012) are shown as well.

| | **D083 (NOAA)** | | **D074 (CHMI)** | | **D064 (DWD)** | | **BP** |
|---|---|---|---|---|---|---|---|
| **Wavelength pair** | **EAC** | **Rel. diff. %** | **EAC** | **Rel. diff. %** | **EAC** | **Rel. diff. %** | **nominal** |
| **A** | 1.788 | 1.03 | 1.7795 | 1.49 | 1.7874 | 1.04 | 1.806 |
| **C** | 0.827 | 0.79 | 0.8224 | 1.29 | 0.8225 | 1.28 | 0.833 |
| **D** | 0.361 | 3.72 | 0.3614 | 3.50 | 0.3568 | 4.82 | 0.374 |
| **AD** | 1.427 | **0.35** | 1.4181 | **0.98** | 1.4306 | **0.10** | 1.432 |
| **CD** | 0.466 | **-1.48** | 0.4610 | **-0.44** | 0.4657 | **-1.50** | 0.459 |

Table 4: Effective Absorption Coefficients (EACs) calculated with IUP cross-sections and relative difference to EACs obtained with "old" Bass/Paur cross-sections and nominal slit functions.