# Peer review of "Optical Characterization of Three Reference Dobsons in the ATMOZ Project – Verification of G.M.B. Dobson's Original Specifications"

_Atmospheric Measurement Techniques, 2017_

## Referee Comment (RC1) · Anonymous Referee #2 · 17 Jan 2018

**1   Introduction**

This paper describes the measurements of the slit function for three traveling reference Dobson instruments. These accurate measurements performed at two National Metrology Institutes are very important to characterize the optical path of these instruments which are transferring to the stations the calibration scale of the global Dobson network. A systematic measure of the Dobson at the different stations could potentially improve the homogeneity of the network as well as improving the accuracy and precision estimations for the Dobson instruments data.
This paper presents interesting and relevant pieces of information within the scope of

[Figure]

AMT.

**2  General comments**

- The text is very "Dobson-Brewer community" oriented with the risk of reducing the impact of this paper. I suggests to the authors to step back a little bit from their research field and adapt parts of the manuscript to gain the interest of a larger community for these interesting results. The absence of a simple form of the total ozone column calculation equation is indicative of this "community approach".

- A comparison with earlier measurements from Komhyr (1993) and Evans (2012) mentioned at lines 17-18 of the introduction is missing.

- The discussion of some results is not precise enough. Sentences like "...differences ... are not too large..." (abstract line 23) or " ...deviate more or less significant ..." (section 3.2 line 25) need to be quantified.

- The three figures 7 a), b) and c) are very similar considering the extended wavelength range. Even though from different instruments, Fig 7b) and c) do not add much information. A plot of the ozone cross-section as an alternative to fig 7 b) would add useful information to the reader. Another possibility would be adding the cross-section to Fig. 4 to 6 to emphasise the importance of the slit function.

- A comment on the consequence (if any !) of having instrument specific EACs on the calibration procedure should be added.

- The paper refers to "Dobson original specifications" for the Dobson slits functions (e.g. first column of table 1). The present ozone absorption coefficients in use in the Dobson network (A : 1.806, C : 0.833, D : 0.374) were first reported in Komhyr et al. (1993) and confirm by Bernhard et al. (2005). The slits functions

used in these two publications for their calculations were indeed those measured by Komhyr et al. as reported in table 1 of Bernhard et al. Please comment on your choice to refer primarily to the "original specifications" and not to the measurements of Komhyr et al. (1993) which are much closer to the measurements presented in this paper.

**3  Specific suggestions**

**Abstract:**

- - lines 15-17: The presence of multiple parenthesis is annoying ! These details should be moved in the main text as they are already partly replicated in the first paragraph of page 3.

- line 23: ... D-wavelength pair.

- line 24: ... ones are not too large in ... : what would be considered as large or small ? please quantify

- line 25: ... to improvements of significantly less ... : is it straightforward being an improvement or just a change ?

**Introduction:**

- line 31: ... stations ( ...

- line 6: Evans (2008) not in the list of references

- line 6: "Fundamental constants" is not the appropriate word if these are instrument dependent.

- line 8: ETCs are determined by comparison with reference instruments. What about the ETC of the references ? A sentence on the absolute calibration procedure is needed here.

- line 24: Köhler (1988) not in the list of references

- line 33: ... have been partly traced back to uncharacterised instrumental features ... : is it a hypothesis ? If not, adding a reference would be useful.

**Measurement procedures in the laboratories**

- line 30: change Fig. 3 to Fig. 2 and Fig. 4 to Fig. 3

- line 31: ...F number equal to #f/4,5 ... : is this information useful ? If yes, explain this notation.

- line 4: vertical plane

- line 11: ... The measured signals were processed ... : which signal, how is it processed ?

**Results:**

- (page 4) line 31: ... Daumont, Brion and Malicet ... : reference needed.

- (page 5) line 30: ... EACs already defined in the abstract.

- (page 6) line 6 : ... less than $\pm 1\%$

**Summary**

- line 20 : ... less than $\pm1\%$

  **Reference**

- Paur and Bass : this reference is not in the alphabetic order

  **Figure:**

- Figure 2: remove the symbols ¶

---

## Referee Comment (RC2) · Anonymous Referee #1 · 19 Jan 2018

General comments

This manuscript reports on very important work that has been long-awaited by the Dobson community. Accurate characterisation of the optical properties of the Dobson instrument by specialist metrological laboratories offers the potential to improve the Dobson algorithm and more accurately determine the uncertainty of ozone measurements made by Dobsons and the consistency of the global network. It is certainly well within the scope of AMT.

I see the two key questions the community would be hoping such an investigation could answer as being:

[Figure]

- What difference to total ozone values do the measured departures from the assumed slit functions make? (Is the difference negligible?)

- How different are individual Dobsons to each other in this regard? Could an overall correction be applied across the whole network, or is there too much variation?

I note that a significant portion of this manuscript has already appeared in an article in the ATMOZ edition of UV News (edition 11 2016) by a subset of five of the current authors. There, the focus was more on the details of the laboratory measurement.

This paper tries to have a somewhat different focus by concentrating on the impact of the laboratory characterisation on ozone measurements and thereby on the Dobson network.

Therefore, my major recommendation is that section 3.2 "Implications of the new effective absorption coefficients" be expanded and made much clearer. The values of the changes given in Tables 3 and 4 are large enough to be worthy of discussion (most notably 0.98% for D074 AD wavelengths calculated with Serdyuchenko), as well the differences between the three instruments (eg CD corrections in Table 3 differ from each other by $\sim$1%). If the three reference instruments differ by the amounts given, wouldn't field instruments be expected to be at least this divergent?

Secondly, in a number of places the statements should be made more precise and quantitative.

Thirdly, there were a large number of small mistakes in English in the text, which could have been removed with more careful proof-reading.

Fourthly, I don't think the paper is clear enough whether it is actually Dobson's manuals that are being referred to or in fact later work (such as Komhyr's), for example in Table 3 the coefficients are called "Dobson/Komhyr" but these are not the values that appear in Dobson 1957. The authors should be accurate about which points go back to Dobson and which do not.

Specific comments

The wording within the parentheses is awkward to read, especially "primary=world standard"

The name of the ATMOZ project is not given correctly

19-23 These sentences should be made more quantitative – the abstract should include the key numbers

I think this statement is very misleading because the difference in AD is up to 0.98% with Bremen cross-sections.

I can't find any example given later on of how the new information could be used to explain unusual results in field calibrations – this seems to be just speculation?

More importantly, if the statement in the abstract is correct, does this imply that field instruments might be significantly different from the reference Dobsons if they were to also be characterised? This should be discussed explicitly. The implication is that field instruments might be quite different and that is why the TuPS is needed. I would like to see this stated clearly.

TOC is not defined.

1920ties should be 1920s.

station should be stations

– 1960ties -> 1960s

– Should be "The fundamental constants"

– My impression of this is that while the assumption that all Dobsons share the same coefficients is clearly a simplification, the effect was always assumed to be relatively small considering other approximations in the algorithm, such as fixed stratospheric temperature and linear height of the ozone layer and many others. If the authors have a similar view this is worth explaining to the reader who otherwise might be puzzled by it.

1970ties/1980ties -> "1970s and early 1980s"

"valid" would be better said to be "prescribed" or a similar word, valid implies they were physically correct.

The term "Effective" is not explained

27-30 These statements should be made quantitative. How much of the difference was claimed to be explained in this way by the studies listed? In particular you should state the findings of Redondas et al., being the most recent work, as it represents the current understanding of the Dobson & Brewer community, and is often referred to.

should be "has offered _an_ opportunity"

A problem with section 2 is that 2.1 and 2.2 appear to have been written quite independently of each other and simply inserted into the document. The writing styles are noticeably different but more importantly, there is no discussion of the similarities and differences between the method and equipment used in the two laboratories. You should add this.

Page 3 line 29 to Page 4 line 14 There are a number of minor English errors which should be corrected in 2.2.

The notation is not quite right because S is written here as a function of lambda and lambda-prime, but then later S is just a function of one wavelength and not two – if you are integrating over all lambda you need to say what lambda-prime is.

4-6 I think this is a very good idea to show results using both the old and new cross-sections!

It would be good to include the Dobson equation to show the relation between total ozone and the absorption coefficients

"bandpass/slit functions" is awkward terminology, it would be better to choose your preferred terms at the start and use them consistently from then on.

Deviations of the "central" wavelength?

Why has the threshold of $\pm 0.05$ nm been chosen as being a "good optical alignment"?

This statement ("With this knowledge . . ." ) is not very clear

Is CD really only a "minor" data set? I am not sure of the proportion of CD observations in the WOUDC but I would have thought there would be a lot given AD-DS is officially restricted to mu < 3 .

This seems to contradict Redondas et al which said the difference between AD and CD was due to the cross-sections, not the slit functions. Please clarify.

Please reword this section to be clearer about what has really been shown – ie how much of the difference between AD and CD can be shown to be the cross-sections and how much the slit functions?

This reads to me to mean the Dobson – Brewer difference will be reduced because of the new cross-sections, but not because of the slit functions – is that what you mean? Later on down the page (Line 19) you say it is the slit functions.

It is implied here that Dobson data should or could be re-evaluated to use these improved slit functions, but how would you actually do it, given the three Dobsons here had differences ranging from -0.310 % to 0.559 % ? (Table 3)

I don't see how you know that the Dobson-Brewer difference will be reduced?

You say less than 1%, but there is a value of 0.98% in Table 4, so wouldn't it be likely that some Dobsons in the network would be greater than 1% ?

In my opinion, the writing here is too colourful for a scientific paper ("perfect", "optimistic prognosis").

You should explain how an intercomparison campaign can confirm that the AD TOC values only need correcting by less than 1%.

This seems to me to be the real conclusion of the paper, that if the TuPS can be used to characterise each individual Dobson in the network, the overall quality of Dobson data will be improved by a noticeable amount. If this is also the view of the authors it should be stated more explicitly.

Evans et al 2012 – does this article exist? I can't find it. Are you able to give a more specific reference?

Figure 2 Carriage return symbols that appear in the diagram should be removed

Figure 7 I would suggest replacing this figure – by eye the reader can't see much difference between the three instruments and we have already seen figures 4, 5 and 6

German-style quotation marks in the table caption

---

## Author Comment (AC1) · 8 Feb 2018

Response on the Reviews ATMOZ-Paper Köhler et al., AMT 2017-411

Referee 1:

Response to General Comments:

- The differences in total ozone are given in Table 3 (comparison between Bass/Paur-EAC and Bass/Paur nominal) and in Table 4 (comparison between IUP-EAC and Bass/Paur nominal). Description in the text on page 5 and 6.

- A separate publication will be written about a series of TuPS-measurements of more

than 10 Dobson and their comparison.

- The references are amended in 2.1. with another Nevas et al., 2016 – publication.

- Expansion of section 3.2.: separate publications planned and mentioned in the text.

- More precise and quantitative statements are included.

- English improvements: Some of them are hopefully corrected applying referees' recommendations. The main author hoped that the review of the original version by one of the native English speakers would have removed most of the improper English wording and grammar.

- Dobson/Komhyr = Dobson slit function + Komhyr Bass/Paur x-sections/absorption coefficients. Dobson used older cross sections, which were valid in the fifties. Komhyr applied adjusted Bass/Paur x-sections using Dobson's nominal slit functions to determine the best set of absorption coefficients.

Special comments:

- P1 l16: Primary = world replaced by only world, locations removed, only countries mentioned - done

- P1 l18: ATMOZ "Traceability for atmospheric total ozone column" – done

- P1 l19-23: numbers mentioned, additionally better description Dobson nominal optical parameters and measured values – done

- P1 l25: better differentiaton between the IUP and BP-results with respect of the 0.98%-difference of D074 in IUP-EACs is done.

- P1 l28: the statement is "it will be possible to explain" (indeed a speculation, but very likely), thus this has still to be investigated. – corresponding amendment done.

- P1, l29: TOC was defined in the abstract (P1, line 24), but not clearly marked – done.

- P1, l29: 1920tie to 1920s – done.
- P1, l30: station to stations – done.

- P2, l3: 1960ties to 1960s – done.

- P2, l6: Fundamental constants is replaced by The essential constants (according referee 2) - done.

- P2, l10: It has been tried to explain the different error sources and their influence a little bit more in detail. In addition the Basher-report has been added to the references.

- P2, l12: 1970s and early 1980s – done

- P2, l18: I think "prescribed" is not better than "valid", I replaced it by "recommended"

- P2, l25: Description of "Effective" is included.

- P2, l27-30: Effect of Teff is quantified as approx. 1%/10K. The statements in Redondas 2014 and Kerr 2002 are a little bit contradictory. A table in Redondas cites only a calculated T-dependence of 0.094%/K for BR#014 in Kerr's paper, whereas Kerr gives in addition a revised value of -0.005%/K. The second one is as far as I know used for elimination of the annual course of the Brewer-Dobson difference, therefore I mentioned this 1%-dependence.

- P3, l3: correction "an" done.

- P3, l8: I agree that the two sections 2.1 and 2.2 are contribution of tow co-authors with different styles of writing. I am not sure whether a rewriting by the main author in order to achieve a "one-style-paper" would be an improvement, as I am not an expert in metrological measurements. The requested discussion of the similarities/differences is added before section 2.1

- P3 – P4: the inconsistent wording "characterisation" has been corrected to "characterization" in the entire text. In addition it was tried to improve the criticized minor English errors.

- P4, l26: corrected.

- P5, l4-6: Thanks for the positive comment.

- P5, l7: Dobson equation and explanation of the EAC effect on TOC included.

- P5, l10 and following: bandpass replaced by slit function in the entire text except under section 2 and 2.1 when this term is referred to the characteristic of the laser beam.

- P5, l17: In my opinion the amendment "central" does not make the content clearer, thus I did not add it here and later in the text.

- P5, l20: Explanation is given that the accepted misalignment of 0.3o of the Q-levers result in the mentioned 0.05 nm. In addition the function of the mentioned Q-levers is referred to the relevant Dobson manuals (Evans 2008 is added).

- P5, l25: More detailed explanation is given.

- P6, l4: Unfortunately I hadn't the occasion and time to find out how many data sets out of almost 1 Mio Dobson data in the WOUDC are CD-based. My long term experience, however, with the European and African Dobsons is, that low and moderate latitude stations normally provide only AD-values as the more accurate data, because even in winter season mu-values below 3.0 or 3.2 (our limit at Hohenpeissenberg) are reached. CD-values come from higher latitude stations like Potsdam/Lindenberg, Hradec Kralove etc only during winter season. Thus my estimation is not completely wrong, that less than 20% (minority!) or even down to 10% of the WOUDC TOC data are CD-observations.

- P6, l7: The findings here are not in contradiction to Redondas et al., they are an amendment: On the one hand one fraction of the AD-CD difference can be explained by the new EACs, but on the other hand the new IUP cross sections can explain another fraction too. Only the cross section effect could be investigated in Redondas et al..

- P6, l10 and l12: This section has been improved (hopefully) to clarify/quantify the effects of EACs and IUP cross sections on the AD-CD Dobson differences and the Dobson-Brewer differences.

- P6, l17: It was clarified that here the re-evaluation is only applied to the reference instruments.

- P6, l19: see under P6, l10 and l12.

- P6, l20: This value refers only to the result of the three standard Dobsons, presented here (see alos P6, l17).

- P6, l23: "perfect" replaced by "very good" and "optimistic" removed.

- P6, l23: The last two sentences of this section are moved in front of the preceding sentence, which makes the context clearer.

- P6, l30: The statement about the TuPS is not a conclusion, but a kind of outlook, to describe the future of Dobson calibrations

- P7, l30: You are right! This publication is hard to find. There is a reference given under the link https://library.wmo.int/opac/index.php?lvl=author_see&id=11665, but when one tries to find it there: no chance. Another link http://www.tandfonline.com/toc/tato20/53/1?nav=tocList was more promising, but no Evans et al. proceeding could be found there as well. Thus I refer now to the corresponding poster, which was presented at the Quadrennial Ozone Symposium 2012 in Toronto and is available from the authors. It is clear, that it is not a peer reviewed publication, but this is the only source we have available.

- P7, Figure 2: carriage return symbols removed

- P13, Figure 7: In contrast to the comment of referee 1 would like to keep this figure in the paper, however, it might be better to show it as an overview first and then the other figures in detail. Thus I moved it as figure 4a-c in front of the detailed figures.

- P15, l5: quotation marks corrected.

Please also note the supplement to this comment:
https://www.atmos-meas-tech-discuss.net/amt-2017-411/amt-2017-411-AC1-supplement.pdf

---

## Author Comment (AC2) · 8 Feb 2018

Response on the Reviews ATMOZ-Paper Köhler et al., AMT 2017-411

Referee 2:

Response to General Comments:

- The basic formula how to calculate ozone is added. I am aware that it is really a specific Dobson-Brewer oriented paper in the context of the ATMOZ project, which is already reflected in the title. I am not sure, how a larger community can be reached with additional or modified parts. Perhaps the addition of "Consequently the quality of the

[Figure]

Dobson TOC records in the data centres will be improved as well, which will increase the reliability of these data for their use in trend analyses and satellite validations" at the end of the second last section of the Summary might somewhat help.

- It is not the intention of the paper to compare the results of former laboratory investigations with the results here. This paper concentrates on the optical characterizations of three reference instruments and wants to show, how large the differences and the effects on the data will be. However, two sentences at the end of 3.2. are added to show the similarity to Evans et al. results.

- More precise and quantitative statements are included in section 3.2 also according referee 1's General comments.

- Figures 7a-c: See also my response to referee 1: "In contrast to the comment of referee 1 would like to keep this figure in the paper, however, it might be better to show it as an overview first and then the other figures in detail. Thus I moved it as figure 4a-c in front of the detailed figures". I don't know, how the presentation of ozone cross sections can enhance the information about the importance of slit functions? Another question would be then: which cross sections? BP or IUP or both together to show their differences and their importance. This would overrun the frame of this paper.

- Consequences of EACs on calibration: See the last sentence in 4. Summary….

- "Dobson original specifications": A similar comment of referee 1 was already answered. One should not mix Dobson slit function with Komhyr Bass/Paur x-sections/absorption coefficients. Dobson used older cross sections, which were valid in the fifties. Komhyr applied adjusted Bass/Paur x-sections using Dobson's nominal slit functions to determine the best set of absorption coefficients.

Special comments:

- P1, l15-17 (abstract): It was tried to remove most of the unneeded parentheses.

- P1, l23: It is not the D-wavelength pair, as suggested, it is the long D-wavelength –
corrected.

- P1, l24: I think the consideration "not too large" is explained in the following sentence by the statement "less than ±1%"

- P1, l25: I think it is indeed an improvement as the data quality will be higher and uncommon behaviour of field Dobsons during calibration can possibly be explained.

- P1, l31 (Introduction): "stations" corrected.

- P2, l6: Missed Evans (2008) reference added.

- P2, l6: - done – see comment under referee 1.

- P2, l8: - done – Langley plot method mentioned.

- P2, l24: done, missing reference added.

- P2, l33: Bernhard et al added as relevant reference.

- P3, l30: Figure numbers corrected.

- P3, l31: This section is a contribution of co-author Smid. I suppose this information is useful.

- P4, l4: plain to plane – done.

- P4, l11: A modified structure of these sentences makes it hopefully clearer, what is meant with "signals" and how they are processed.

- P4, l31: Three relevant references of Daumont, Brion and Malicet have been added

- P5, l30: term "effective absorption coefficients" removed.

- P6, l6: + replaced by ±.

- P6, l20, Summary: + replaced by ±.

- References: Pass and Bass moved to alphabetically right place

- Figur 2: Symbols removed.

Please also note the supplement to this comment:
https://www.atmos-meas-tech-discuss.net/amt-2017-411/amt-2017-411-AC2-supplement.pdf
* * *

---

## Author Response (AR1)

**Response on the Reviews ATMOZ-Paper Köhler et al., AMT 2017-411**

**Referee 1:**

10

15

20

Response to General Comments:

- The differences in total ozone are given in Table 3 (comparison between Bass/Paur-EAC and Bass/Paur nominal) and in Table 4 (comparison between IUP-EAC and Bass/Paur nominal).
   Description in the text on page 5 and 6.
  - A separate publication will be written about a series of TuPS-measurements of more than 10 Dobson and their comparison.
  - The references are amended in 2.1. with another Nevas et al., 2016 publication.
  - Expansion of section 3.2.: separate publications planned and mentioned in the text.
    - More precise and quantitative statements are included.
    - English improvements: Some of them are hopefully corrected applying referees'
    - recommendations. The main author hoped that the review of the original version by one of the native English speakers would have removed most of the improper English wording and grammar.
      - Dobson/Komhyr = Dobson slit function + Komhyr Bass/Paur x-sections/absorption coefficients.
         Dobson used older cross sections, which were valid in the fifties. Komhyr applied adjusted
         Bass/Paur x-sections using Dobson's nominal slit functions to determine the best set of
  - absorption coefficients.
  - Special comments:
    - P1 I16: Primary = world replaced by only world, locations removed, only countries mentioned done
    - P1 I18: ATMOZ "Traceability for atmospheric total ozone column" done
- P1 I19-23: numbers mentioned, additionally better description Dobson nominal optical parameters and measured values – done
  - P1 I25: better differentiaton between the IUP and BP-results with respect of the 0.98%difference of D074 in IUP-EACs is done.
- P1 I28: the statement is "it will be possible to explain" (indeed a speculation, but very likely),
   thus this has still to be investigated. corresponding amendment done.
  - P1, I29: TOC was defined in the abstract (P1, line 24), but not clearly marked done.
    - P1, I29: 1920tie to 1920s done.
    - P1, I30: station to stations done.
  - P2, I3: 1960ties to 1960s done.
- P2, I6: Fundamental constants is replaced by The essential constants (according referee 2) done.
  - 1

- P2, I10: It has been tried to explain the different error sources and their influence a little bit more in detail. In addition the Basher-report has been added to the references.
- P2, I12: 1970s and early 1980s done
- P2, I18: I think "prescribed" is not better than "valid", I replaced it by "recommended"
- P2, I25: Description of "Effective" is included.
- P2, I27-30: Effect of Teff is quantified as approx. 1%/10K. The statements in Redondas 2014 and Kerr 2002 are a little bit contradictory. A table in Redondas cites only a calculated T-dependence of 0.094%/K for BR#014 in Kerr's paper, whereas Kerr gives in addition a revised value of -0.005%/K. The second one is as far as I know used for elimination of the annual course of the Brewer-Dobson difference, therefore I mentioned this 1%-dependence.
- P3, I3: correction "an" done.

10

15

25

- P3, I8: I agree that the two sections 2.1 and 2.2 are contribution of tow co-authors with different styles of writing. I am not sure whether a rewriting by the main author in order to achieve a "one-style-paper" would be an improvement, as I am not an expert in metrological measurements. The requested discussion of the similarities/differences is added before section
- 2.1 2.2 D4 the inconsistent wording "shore torienties" has been corrected to "shore starterization"
  - P3 P4: the inconsistent wording "characterisation" has been corrected to "characterization" in the entire text. In addition it was tried to improve the criticized minor English errors.
    P4, I26: corrected.
- 20 P5, I4-6: Thanks for the positive comment.
  - P5, I7: Dobson equation and explanation of the EAC effect on TOC included.
    - P5, I10 and following: bandpass replaced by slit function in the entire text except under section 2 and 2.1 when this term is referred to the characteristic of the laser beam.
  - P5, I17: In my opinion the amendment "central" does not make the content clearer, thus I did not add it here and later in the text.
- P5, I20: Explanation is given that the accepted misalignment of 0.3° of the Q-levers result in the mentioned 0.05 nm. In addition the function of the mentioned Q-levers is referred to the relevant Dobson manuals (Evans 2008 is added).
- P5, I25: More detailed explanation is given.
- P6, I4: Unfortunately I hadn't the occasion and time to find out how many data sets out of almost 1 Mio Dobson data in the WOUDC are CD-based. My long term experience, however, with the European and African Dobsons is, that low and moderate latitude stations normally provide only AD-values as the more accurate data, because even in winter season mu-values below 3.0 or 3.2 (our limit at Hohenpeissenberg) are reached. CD-values come from higher
- 35 latitude stations like Potsdam/Lindenberg, Hradec Kralove etc only during winter season. Thus my estimation is not completely wrong, that less than 20% (minority!) or even down to 10% of the WOUDC TOC data are CD-observations.
  - P6, I7: The findings here are not in contradiction to Redondas et al., they are an amendment: On the one hand one fraction of the AD-CD difference can be explained by the new EACs, but

on the other hand the new IUP cross sections can explain another fraction too. Only the cross section effect could be investigated in Redondas et al..

- P6, I10 and I12: This section has been improved (hopefully) to clarify/quantify the effects of EACs and IUP cross sections on the AD-CD Dobson differences and the Dobson-Brewer differences.
- P6, I17: It was clarified that here the re-evaluation is only applied to the reference instruments.
- P6, I19: see under P6, I10 and I12.

5

25

- P6, I20: This value refers only to the result of the three standard Dobsons, presented here (see alos P6, I17).
- 10 P6, I23: "perfect" replaced by "very good" and "optimistic" removed.
  - P6, I23: The last two sentences of this section are moved in front of the preceding sentence, which makes the context clearer.
    - P6, I30: The statement about the TuPS is not a conclusion, but a kind of outlook, to describe the future of Dobson calibrations
- P7, I30: You are right! This publication is hard to find. There is a reference given under the link <a href="https://library.wmo.int/opac/index.php?lvl=author\_see&id=11665">https://library.wmo.int/opac/index.php?lvl=author\_see&id=11665</a>, but when one tries to find it there: no chance. Another link <a href="http://www.tandfonline.com/toc/tato20/53/1?nav=tocList">https://library.wmo.int/opac/index.php?lvl=author\_see&id=11665</a>, but when one tries to find it there: no chance. Another link <a href="http://www.tandfonline.com/toc/tato20/53/1?nav=tocList">http://www.tandfonline.com/toc/tato20/53/1?nav=tocList</a> was more promising, but no Evans et al. proceeding could be found there as well. Thus I refer now to the corresponding poster, which was presented at the Quadrennial Ozone Symposium 2012</a>
- 20 in Toronto and is available from the authors. It is clear, that it is not a peer reviewed publication, but this is the only source we have available.
  - P7, Figure 2: carriage return symbols removed
  - P13, Figure 7: In contrast to the comment of referee 1 would like to keep this figure in the paper, however, it might be better to show it as an overview first and then the other figures in detail. Thus I moved it as figure 4a-c in front of the detailed figures.
  - P15, I5: quotation marks corrected.

**Response on the Reviews ATMOZ-Paper Köhler et al., AMT 2017-411**

**Referee 2:**

15

30

35

Response to General Comments:

- The basic formula how to calculate ozone is added. I am aware that it is really a specific Dobson-Brewer oriented paper in the context of the ATMOZ project, which is already reflected in the title. I am not sure, how a larger community can be reached with additional or modified parts. Perhaps the addition of "Consequently the quality of the Dobson TOC records in the data centres will be improved as well, which will increase the reliability of these data for their use in trend analyses and satellite validations" at the end of the second last section of the Summary might somewhat help.
  - It is not the intention of the paper to compare the results of former laboratory investigations with the results here. This paper concentrates on the optical characterizations of three reference instruments and wants to show, how large the differences and the effects on the data will be. However, two sentences at the end of 3.2. are added to show the similarity to Evans et al. results.
    - More precise and quantitative statements are included in section 3.2 also according referee 1's General comments.
- Figures 7a-c: See also my response to referee 1: "In contrast to the comment of referee 1 would like to keep this figure in the paper, however, it might be better to show it as an overview first and then the other figures in detail. Thus I moved it as figure 4a-c in front of the detailed figures". I don't know, how the presentation of ozone cross sections can enhance the information about the importance of slit functions? Another question would be then: which cross sections? BP or IUP or both together to show their differences and their importance. This would 25
  - Consequences of EACs on calibration: See the last sentence in 4. Summary....
  - "Dobson original specifications": A similar comment of referee 1 was already answered. One should not mix Dobson slit function with Komhyr Bass/Paur x-sections/absorption coefficients. Dobson used older cross sections, which were valid in the fifties. Komhyr applied adjusted Bass/Paur x-sections using Dobson's nominal slit functions to determine the best set of absorption coefficients.

**Special comments:**

- P1, I15-17 (abstract): It was tried to remove most of the unneeded parentheses.
- P1, I23: It is not the D-wavelength pair, as suggested, it is the long D-wavelength corrected.
  - 4

- P1, I24: I think the consideration "not too large" is explained in the following sentence by the statement "less than ±1%"
- P1, I25: I think it is indeed an improvement as the data quality will be higher and uncommon behaviour of field Dobsons during calibration can possibly be explained.
- P1, I31 (Introduction): "stations" corrected.
- P2, I6: Missed Evans (2008) reference added.
- P2, I6: done see comment under referee 1.
- P2, I8: done Langley plot method mentioned.
- P2, I24: done, missing reference added.
- P2, I33: Bernhard et al added as relevant reference.
  - P3, I30: Figure numbers corrected.
  - P3, I31: This section is a contribution of co-author Smid. I suppose this information is useful.
  - P4, I4: plain to plane done.

10

- P4, I11: A modified structure of these sentences makes it hopefully clearer, what is meant with "signals" and how they are processed.

- P4, I31: Three relevant references of Daumont, Brion and Malicet have been added
- P5, I30: term "effective absorption coefficients" removed.
- P6, I6: + replaced by ±.
- P6, I20, Summary: + replaced by ±.
- 20 References: Pass and Bass moved to alphabetically right place
  - Figur 2: Symbols removed.

**Optical Characterization of Three Reference Dobsons in the ATMOZ Project – Verification of G.M.B. Dobson's Original Specifications**

Ulf Köhler1, Saulius Nevas2, Glen McConville3, Robert Evans3, Marek Smid4, Martin Stanek5, Alberto Redondas6, and Fritz Schönenborn1

1Met. Obs. Hohenpeissenberg, Deutscher Wetterdienst, Albin-Schwaiger-Weg 10, 82383 Hohenpeissenberg, Germany
 2Physikalisch-Technische Bundesanstalt, Bundesallee 100, 38116 Braunschweig, Germany
 3ESRL, NOAA, 325 Broadway, 325 Boulder, USA

4Optical Radiometry and Photometry Dept., Czech4Czech Metrology Institute, V Botanice 4, 150 72 Praha 5 Okruzni 31,
 638 00 Brno, Czech Republic, (Dept. of Optics, Prague)

5Solar and Ozone Observatory, Czech Hydrometeorological Institute, Zamecek 456, 500 08 Hradec Kralove 8, Czech Republic

6Izaña Atmospheric Research Center, AEMET- Meteorological State Agency, C/ La Marina 20, 6 Planta, 38071 Santa Cruz de Tenerife, Spain

15 Correspondence to: Ulf Köhler (ulf.koehler@dwd.de)

5

Abstract. Three reference Dobsons (regional standards Dobsons No. 064 Hohenpeissenberg, Germany and No. 074 Hradec Kralove, Czech Republic and primary = as well as the world standard Dobson-No. 083Boulder, USA) were optically characterized at PTB (the Physikalisch-Technische Bundesanstalt (PTB) in Braunschweig) in 2015 and at CMI (the Czech Metrology Institute (CMI) in Prague) in 2016 within the EMRP ENV 059 project "Traceability for theatmospheric

- 20 total column ozone". BandpassSlit functions and the related parameters of the instruments were measured and compared with G. M. B. Dobson's specification in his handbook. AAll Dobsons show a predominantly good match of the bandpassSlit functions and the peak (centroid) wavelengths of D083, D064with deviations between -0.11 and +0.12 nm and D074 with differences of the Full Width Half Maximum (FWHM) between 0.13nm and 0.37nm compared to the nominal values could be observed at the shorter wavelengths. Slightly larger deviations of the FWHMs from the nominal Dobson data, up to 1.22
- 25 nm, can be seen inat the longer wavelengths, especially infor the slit function of the long D-wavelength. As consequence of these findings the However, differences ofbetween the derived Effective Absorptions Coefficients (EACs) for ozone toderived using Dobson's nominal <del>ones</del>
[revised manuscript text omitted]
 functionfunctions were analysed for errorerrors due to nonzeronon-zero bandwidth of the measuring beam.
- 25 And itIt turned out that there was no need for any correction forof the used 0.1 nm FWHM slit-width-used.

**3 Results**

**3.1. Cross-sections, slit functions and effective absorption coefficients (EACs)**

The derivation of the EACs for each individual Dobson (using the specific slit functions  $S(\lambda)$  measured in the laboratories) is described in detail in Bernhard et al., 2005 and Redondas et al., 2014. For this calculation the following approximate Eq. (1)

30 is used:

|------------------------------------------------------------------------------------------------------|-------------------------------------|
| $\int \sigma(\lambda) S_i(\lambda, \lambda') d\lambda$                                               | Formatiert: Schriftartfarbe: Text 1 |
| Eq (1) $\alpha_i = \frac{\int \delta(\lambda) S_i(\lambda) d\lambda}{\int \sigma(\lambda) d\lambda}$ |                                     |
| Eq (1) $\alpha_i = \frac{1}{\int S_i(\lambda) d\lambda}$                                             |                                     |

**5 where**

 $\rho(\lambda)$  is the ozone cross-section for the corresponding wavelength at the fixed temperature of -46.3° C for the Dobson network (after Bass and Paur since 1992 and after IUP in the future).

 $S_{\Sigma_i}(Q)$  is the measured instrument slit function for the corresponding wavelength.  $\alpha_{\alpha_i}$  is the approximate approximated effective absorption coefficient EAC.

**10**

15

Since 1992 the The above-mentioned cross-sections after Bass and Paur have been in use, but recently since 1992. However, the International Ozone Commission decided recently to replace these old cross-sections by new ones. After the first proposal to use the results derived from **D**aumont, **B**rion and **M**alicet (DBM, Daumont et al., 1992, Brion et al., 1993 and Malicet et al., 1995) it was found by Redondas et al., 2014, that the IUP ozone cross-sections, determined at the University of Bremen, Institute of Experimental Physics (IUP) (Gorshelev et al., 2013; Serdyuchenko, 2013), give a much better agreement of the TOC measured with by Dobsons and Brewers, respectively. The introduction of these IUP cross-sections into the global network is finally decided, but not completed yet.

To get a complete picture of the impact of using the effective ozone absorption coefficients, it was decided to compare not only the various sets (nominal and effective ones) of coefficients after Bass and Paur, but also to include the TOC-values in this comparison, derived using individual Dobson EACs derived withbased on the new set of IUP absorption cross-sections. It is a very simple, almost direct correlation between the TOC values and the variation of the EACs, apparent when looking to the general ozone calculation formula for the single wavelength pair:

25
$$\frac{\text{Eq (2)}}{(\alpha - \alpha')\mu} X = \frac{[N - (\beta - \beta')\frac{mp}{p_0} - (\delta - \delta') \sec(SZA)]}{(\alpha - \alpha')\mu}$$

where

$$X = \text{total amount of ozone expressed in Dobson Units (1 DU = 10-5 m pure ozone at STP), or in atmo-cm;}$$

$$N = L_0 - L = \log(I_0 / I'_0) - \log(I / I')$$
30
$$I_0 \text{ and } I_0' = \text{intensities outside the atmosphere of solar radiation at the short and long wavelengths, respectively;}$$

$$I \text{ and } I' = \text{measured intensities of solar radiation at the short and long wavelengths, respectively;}$$

[revised manuscript text omitted]